# Bulk tungsten-substituted vanadium oxide for low-temperature NOx removal in the presence of water

Yusuke Inomata [1], Hiroe Kubota[2], Shinichi Hata[3], Eiji Kiyonaga[4], Keiichiro Morita[4], Kazuhiro Yoshida[4], Norihito Sakaguchi [5], Takashi Toyao [2], Ken-ichi Shimizu[2,8], Satoshi Ishikawa[6], Wataru Ueda[6], Masatake Haruta[1] & Toru Murayama [1,7,8 ✉]

NH$_3$-SCR (selective catalytic reduction) is important process for removal of NOx. However, water vapor included in exhaust gases critically inhibits the reaction in a low temperature range. Here, we report bulk W-substituted vanadium oxide catalysts for NH$_3$-SCR at a low temperature (100–150 °C) and in the presence of water (~20 vol%). The 3.5 mol% W-substituted vanadium oxide shows >99% (dry) and ~93% (wet, 5–20 vol% water) NO conversion at 150 °C (250 ppm NO, 250 ppm NH$_3$, 4% O$_2$, SV = 40000 mL h$^{-1}$ g$_{cat}$$^{-1}$). Lewis acid sites of W-substituted vanadium oxide are converted to Brønsted acid sites under a wet condition while the distribution of Brønsted and Lewis acid sites does not change without tungsten. NH$_4$$^+$ species adsorbed on Brønsted acid sites react with NO accompanied by the reduction of V$^{5+}$ sites at 150 °C. The high redox ability and reactivity of Brønsted acid sites are observed for bulk W-substituted vanadium oxide at a low temperature in the presence of water, and thus the catalytic cycle is less affected by water vapor.

[1] Research Center for Gold Chemistry, Graduate School of Urban Environmental Sciences, Tokyo Metropolitan University Hachioji, Tokyo 192-0397, Japan. [2] Institute for Catalysis, Hokkaido University, Sapporo, Hokkaido 001-0021, Japan. [3] Department of Applied Chemistry, Faculty of Engineering, Sanyo-Onoda City University, Sanyo-Onoda, Yamaguchi 756-0884, Japan. [4] Energia Economic and Technical Research Institute, The Chugoku Electric Power Company, Incorporated, Higashihiroshima, Hiroshima 739-0046, Japan. [5] Laboratory of Integrated Function Materials, Center for Advanced Research of Energy and Materials, Faculty of Engineering, Hokkaido University, Sapporo, Hokkaido 060-8628, Japan. [6] Department of Material and Life Chemistry, Faculty of Engineering, Kanagawa University, Yokohama, Kanagawa 221-8686, Japan. [7] Yantai Key Laboratory of Gold Catalysis and Engineering, Shandong Applied Research Center of Gold Nanotechnology (Au-SDARC) School of Chemistry and Chemical Engineering, Yantai University, Yantai 264005, China. [8] These authors jointly supervised this work: Ken-ichi Shimizu, Toru Murayama. ✉email: murayama@tmu.ac.jp

Emission control of $NO_x$ (NO and $NO_2$) is an important task for an industrial chemical process that relies on thermal energy produced by the combustion of fossil fuels. For stationary $deNO_x$ systems such as those in coal-fired power plants and waste treatment plants, selective catalytic reduction (SCR) has been used with ammonia as a reducing reagent to convert harmful $NO_x$ to harmless $N_2$ and $H_2O$[1–3]:

$$4NO + 4NH_3 + O_2 \rightarrow 4N_2 + 6H_2O. \qquad (1)$$

Although the stationary $deNO_x$ systems have been successfully established, a conventional catalyst needs a high temperature (>300 °C) for $NH_3$-SCR to proceed[3,4]. In most cases, $deNO_x$ catalysts are placed just after the boiler system for a high reaction temperature[4]. Consequently, $deNO_x$ catalysts are deactivated by ash and sulfate generated from the reaction of ammonia and $SO_2$ because the catalysts are exposed to the gas before passing through the dust collection and $deSO_x$ system. $DeNO_x$ catalysts can be moved to the latter part of the system to avoid the deactivation of the $deNO_x$ catalyst works at a low temperature (<150 °C) since the downstream gas is usually cooled down to 100–150 °C.

Vanadium (V) oxide-based catalysts ($V_2O_5/TiO_2$, $V_2O_5$–$WO_3$/$TiO_2$) have been used as industrial catalysts for stationary boiler systems because they show high $N_2$ selectivity, good thermal stability, and low $SO_2$ oxidation activity to $SO_3$[2]. However, their high working temperature limits the latitude of the $deNO_x$ process. Therefore, many studies have been carried out to develop low-temperature SCR catalysts such as metal oxide-based materials (V, Mn, Cr, Cu, W, Ce, and Fe)[1,5–10] and ion-exchanged zeolite (Cu-ZSM-5, Fe-ZSM-5, Cu-CHA, and Cu-SSZ-13)[1,11–17].

We previously showed that vanadium oxide with a bulk crystal structure works as an $NH_3$-SCR catalyst at a low temperature (<150 °C) with high $N_2$ selectivity and less $SO_2$ oxidation activity in contrast to the supported vanadia species[18,19]. Recent studies provided an insight into the effect of the coordination environment, atomic configuration, and vanadium surface density of the catalyst on SCR activity[20–22]. It was proposed in previous reports that supported vanadia catalysts are composed of $VO_4$ units with O-exposed moieties[2,23]. On the other hand, bulk $V_2O_5$ has a metal (V)-exposed part in addition to the O-exposed part[24], and it might show a different catalytic property from that of a supported vanadia catalyst because of their different redox properties, bond strengths, and coordination environments[23,25]. Bulk crystal of $V_2O_5$ is composed of a two-dimensional sheet structure with $VO_5$ units that are connected by a weak van der Waals force[24,26–29]. Consequently, the morphology of $V_2O_5$ is thermally reconstructed because of its weak interaction in the crystal structure. According to previous studies, the poor stability of $V_2O_5$ can be overcome by doping another metal with a three-dimensional coordination environment[30–32]. Bulk tungsten oxide ($WO_3$) is composed of $WO_6$ units that are three-dimensionally connected[33–35]. Furthermore, $WO_3$ is known as a promoter for supported vanadia catalysts that enhance the reactivity of vanadia sites[36]. Therefore, the incorporation of tungsten sites into bulk $V_2O_5$ would facilitate not only the $NH_3$-SCR activity but also the structural stability of bulk vanadium oxide-based catalysts.

The negative effect of water, which is intrinsically included in the exhaust gas (10–30 vol%)[2], is critical for low-temperature $NH_3$-SCR, while the effect would be much less in a high-temperature range. The effect of water is thought to be inhibition of the adsorption of the reactant[37–43] and/or the reaction between NO and adsorbed $NH_3$[44–46]. The water adsorption behavior was studied for a $V_2O_5$–$WO_3$ solid solution system and $V_2O_5$ without tungsten[47–49]. Broclawik et al.[48] theoretically suggested that dissociative adsorption of water occurs on adjacent V–O–W sites, leading to an increase in the concentration of Brønsted acid sites.

On the other hand, water merely adsorbs in the case of $V_2O_5$ without tungsten, and a Brønsted acid site is not newly created. Thus, $NH_3$-SCR would efficiently proceed with a vanadium–tungsten complex oxide even in the presence of water vapor owing to the large population of protonic Brønsted acid sites.

The reaction mechanisms of $NH_3$-SCR have been widely investigated for a supported vanadium-based oxide catalyst under a dry gas condition using in situ and *operando* spectroscopic techniques (infrared (IR), ultraviolet–visible (UV–Vis), Raman)[20,36,38,46,50–54]. The current consensus is that both Lewis acid and Brønsted acid sites participate in the reaction mechanism. The reaction mechanism should be investigated in the presence of water as well as under dry conditions in order to understand catalytic activity in the actual atmosphere. Although Topsøe et al.[45] and Song et al.[55] investigated the reaction mechanism of 3–6 wt% $V_2O_5/TiO_2$ under a 1.7–3.0 vol% water atmosphere at 250–300 °C, the reaction mechanism of the vanadium–$WO_3$ system has not been investigated under wet conditions, especially in a low-temperature range. An insight into the reaction mechanism under wet conditions would provide a milestone for designing the active site of a low-temperature $NH_3$-SCR catalyst under actual conditions.

Herein, we report a bulk tungsten-substituted vanadium oxide catalyst for low-temperature $NH_3$-SCR (<150 °C) under a wet atmosphere. The structural features of bulk tungsten-substituted vanadium oxide are studied. We examine the effect of tungsten substitution on water tolerance to $NH_3$-SCR activity and catalytic stability. The reaction mechanism is investigated by *operando* spectroscopies under dry and wet conditions to elucidate the role of vanadium and tungsten sites in the catalytic cycle. Finally, we compare the difference between bulk tungsten-substituted vanadium oxide catalyst and a current titania-supported vanadia catalyst.

## Results

**Synthesis of tungsten-substituted vanadium oxide**. Tungsten-substituted vanadium oxide was synthesized by the oxalate method using ammonia metavanadate, ammonia metatungstate, and oxalic acid. Water-insoluble ammonia metavanadate was converted to soluble vanadium oxalate in aqueous media, and ammonia metatungstate was then added to the aqueous solution followed by evaporation. The resulting precursor powder was then calcined in air. The synthesized $x$ mol% tungsten-substituted vanadium oxides were denoted as $x$W–V ($x$ = 0, 1, 3.5, 7, 10, 15, and 40). The precursor powder of the 3.5W–V sample showed a strong exothermic peak assigned to its decomposition at around 297 °C from thermogravimetry/differential thermal analysis measurement (Supplementary Fig. 2). Thus, we calcined the catalyst twice at 300 °C for 4 h each time to obtain $V_2O_5$-based catalysts with large surface areas. The specific surface areas of the $x$W–V ($x$ = 0–40) catalysts were measured to be 32–41 $m^2 g^{-1}$ (Supplementary Table 2), which are almost in the same range and comparably large.

**Characterization of tungsten-substituted vanadium oxide**. Atomic-resolution high-angle annular dark-field imaging-scanning transmission electron microscopy (HAADF-STEM) images were measured to directly determine the incorporation of tungsten into vanadium oxide. The image of 0W–V ($V_2O_5$ without tungsten) showed a lattice fringe based on the (101) plane (Fig. 1a). No bright spots of tungsten atoms were observed for 0W–V. On the other hand, we confirmed atomic bright spots on the vanadium oxide lattice for 3.5W–V (Fig. 1b), indicating that tungsten atoms were atomically dispersed. An enlarged view of

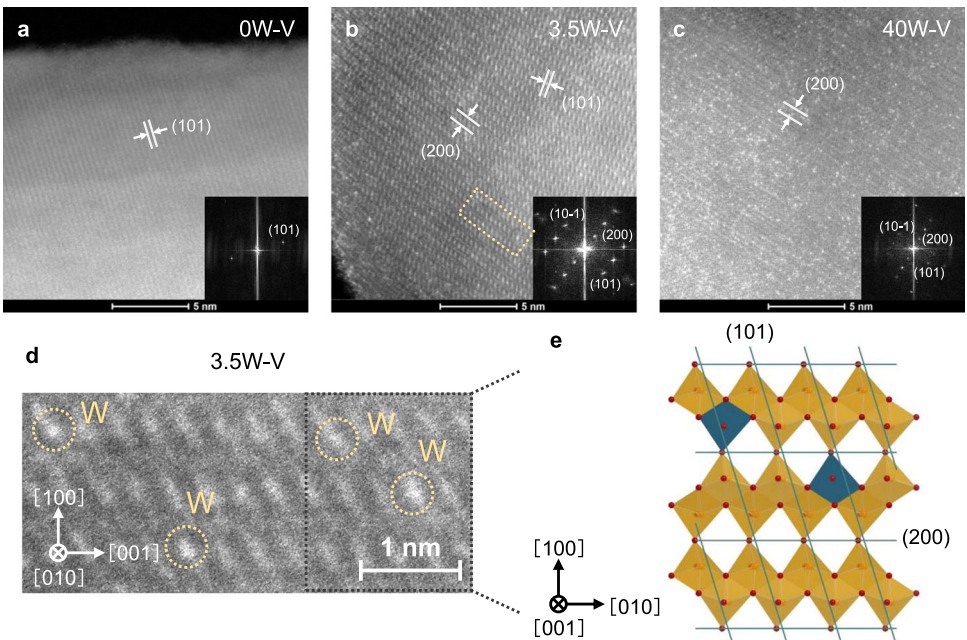

**Fig. 1 Direct observation of tungsten substitution.** HAADF-STEM images of **a** 0, **b** 3.5, and **c** 40 mol% W-substituted vanadium oxide (0, 3.5, and 40W–V). Insets: corresponding FFT images. **d** Enlarged view of the yellow region in (**b**). **e** Crystal structure of W-substituted vanadium oxide. Orange polyhedra: vanadium units; blue polyhedra: tungsten units.

the HAADF-STEM image for 3.5W–V (Fig. 1d) showed that lattice vanadium atoms were partially substituted by tungsten atoms (Fig. 1e). Aggregated tungsten sites were observed on the surfaces of catalysts when 7 mol% tungsten was added to vanadium oxide (7W–V; Supplementary Fig. 3). Adjacent and aggregated tungsten moieties were more clearly observed when 40 mol% tungsten was added to vanadium oxide (40W–V; Fig. 1c), indicating that a $WO_3$ phase partially forms with an increase in the amount of tungsten. Thus, 0, 3.5, and 40W–V can be regarded as catalysts without tungsten, with a moderate amount of tungsten and with an excess amount of tungsten, respectively.

The crystal structures of $x$W–V were confirmed from X-ray powder diffraction (XRD) measurements to determine the effect of tungsten doping (Fig. 2a). For 0, 1, and 3.5W–V, all of the diffraction peaks were assigned as vanadium oxide phase (Fig. 2a, b), and no $WO_3$ crystalline phase was confirmed. Given that the HAADF-STEM images showed tungsten dispersed atomically, the XRD patterns suggested that tungsten atoms were atomically incorporated into the vanadium oxide lattice and a W–V oxide solid solution was formed. We confirmed broad XRD peaks of $WO_3$ (Fig. 2a, c) at $2\theta = 23°$ for 7, 10, 15, and 40W–V in addition to the XRD patterns of vanadium oxide. These results indicate that tiny $WO_3$ crystalline particles form when the tungsten content is higher than 7 mol% as shown by HAADF-STEM images. The peak positions of (010) reflection at around $2\theta = 20.2°$ slightly increased with an increase in the amount of tungsten (Fig. 2a, inset), indicating that the lattice spacing decreased along the $b$-axis. The lattice parameters were calculated by Rietveld analysis. The lattice parameters of the $b$-axis decreased with an increase in the amount of tungsten (Supplementary Fig. 4b). These results show that $WO_6$ units connect the vanadium oxide layers by bond formation (e.g., V = O → V–O–W). The lattice parameters along the $a$-axis and $c$-axis were expanded because of the incorporation of tungsten atoms with a larger ionic radius (Supplementary Fig. 4a, c).

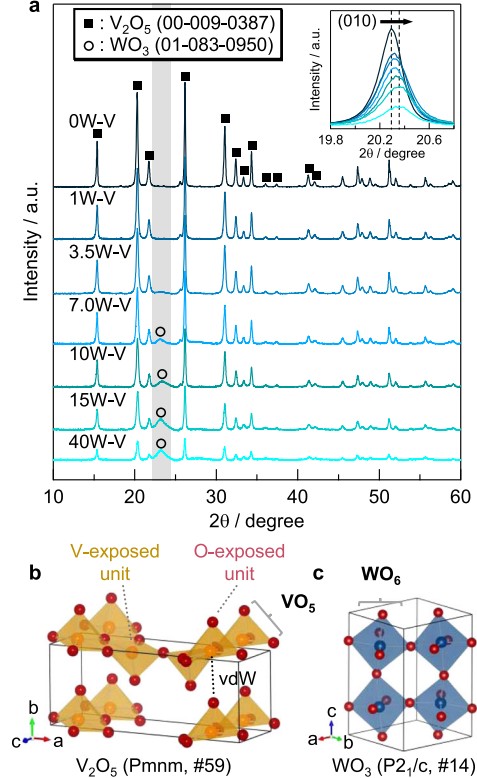

**Fig. 2 Crystal structures of W-substituted vanadium oxide catalysts. a** XRD patterns of W-substituted vanadium oxide catalysts. Inset: enlarged view of peaks corresponding to (010) reflection. (All of the peaks $2\theta > 40°$ were also assigned to $V_2O_5$.) Crystal structures of **b** $V_2O_5$ and **c** $WO_3$. Orange polyhedra: $VO_5$ units; blue polyhedra: $WO_6$ units. VdW van der Waals force.

We also conducted IR measurements for 0, 3.5, and 40W–V to determine the effect of tungsten substitution on the structure (Supplementary Fig. 5). Although an obvious difference in the infrared (IR) spectra was not found for 3.5 and 0W–V, the IR peak of 40W–V assigned to V=O (1020 cm$^{-1}$) became weak compared to those of 0 and 3.5W–V, results similar to those reported by Satsuma et al.[56]. The number of V=O bonds would decrease to form V–O–W bonds by the addition of tungsten into the vanadium oxide lattice.

## NH$_3$-SCR activity of tungsten-substituted vanadium oxide.

First, we investigated the effects of different amounts of tungsten on the NH$_3$-SCR activity of $x$W–V catalysts under dry and wet (10 vol% water) atmospheres at 150 °C (Fig. 3a). The 0W–V catalyst (V$_2$O$_5$ without tungsten) showed NO conversions of 82% (dry) and 47% (wet). The NO conversion increased with an increase in the amount of tungsten up to 3.5 mol%. The NO conversions of 3.5W–V were >99% (dry) and 94% (wet). The negative effect of water was suppressed by the incorporation of tungsten into vanadium oxide. The 3.5W–V catalyst showed the best NH$_3$-SCR activity among the synthesized catalysts, while the NO conversion decreased with a further increase in the amount of tungsten. The 40W–V catalyst showed NO conversions of 29% (dry) and 19% (wet). Thus, an excess amount of tungsten decreased the NH$_3$-SCR activity.

The dependence of NO conversion on reaction temperature was examined for 0W–V, 3.5W–V, and 1 wt% V$_2$O$_5$–5 wt% WO$_3$/TiO$_2$ (V–W/TiO$_2$, a model of an industrial catalyst; Fig. 3b). The 3.5W–V catalyst showed NH$_3$-SCR activity at a low temperature (<150 °C) under both dry and wet conditions in which the conventional

V–W/TiO$_2$ catalyst did not show sufficient NH$_3$-SCR activity. The 0W–V catalyst (V$_2$O$_5$ without tungsten) also showed low-temperature NH$_3$-SCR activity under a dry condition, but the negative effect of water was critical at a low temperature. The temperatures for 80% NO conversion ($T_{80}$) were 110 °C (3.5W–V), 157 °C (0W–V), and 225 °C (V–W/TiO$_2$) under a dry condition and 135 °C (3.5W–V), 178 °C (0W–V), and 236 °C (V–W/TiO$_2$) under a wet (10 vol% water) condition. N$_2$ selectivity was >99% during the reaction, and undesirable N$_2$O production was not observed for any of the catalysts (Supplementary Fig. 6).

The effects of water concentration in flow gas on NH$_3$-SCR were also investigated for 0W–V and 3.5W–V at 150 °C (Fig. 4a). Although NO conversion of 3.5W–V was slightly decreased by the addition of 5% water into reaction gas, ~93% NO conversion was maintained in the presence of 5–20 vol% of water. On the other hand, NO conversion was drastically decreased from 82% to 35% in the case of 0W–V when water vapor was introduced to the reaction gas. Although NO conversion decreased with the addition of water, the values were almost the same regardless of the concentration of water (2–20%). Previous studies showed that a further inhibitory effect of water on the reaction rate did not occur at a high water concentration (>5%)[2,39,57]. Water vapor would affect a catalytic cycle more strongly at a low temperature such as 150 °C. Thus, we found that tungsten-doped vanadium oxide (3.5W–V) shows a low-temperature NH$_3$-SCR activity in the presence of a high concentration of water.

A catalytic stability test (150 °C) was conducted for 0W–V and 3.5W–V to determine the effect of tungsten on the stability (Fig. 4b). In the presence of 10 vol% water, the NO conversion was decreased for both 0W–V and 3.5W–V. After turning off the water addition, the NO conversion of 3.5W–V recovered to the original value (>99%), but that of 0W–V decreased from 78 to 59%. The specific surface areas of 0 and 3.5W–V were measured before and after the catalytic stability test (Fig. 4c). The specific surface area of 0W–V decreased after the stability test (41 → 19 m$^2$ g$^{-1}$), while 3.5W–V had the same value (39 m$^2$ g$^{-1}$). Although a porous structure was confirmed by scanning electron microscope measurement for 0W–V before the activity test, it disappeared and the surface became smooth after the test (Supplementary Fig. 7). Obvious morphological changes were not confirmed for 3.5W–V. There were no changes in the XRD patterns of 0 and 3.5W–V after the stability test (Supplementary Fig. 8). We assume that the bulk tungsten sites, which have a three-dimensional coordination environment (WO$_6$ units), retain the structure by incorporation into the layered vanadium oxide lattice, leading to the retention of surface area.

## Reaction kinetics of tungsten-substituted vanadium oxide catalysts.

Reaction rate, apparent activation energy ($E_a$), and reaction order were investigated to understand the reaction kinetics for low-temperature NH$_3$-SCR. Reaction rates for the kinetics were calculated by adjusting the weights of the catalysts to control the conversion to <20%. Reaction rates per surface area were $3.2 \times 10^{-9}$ (0W–V), $5.2 \times 10^{-9}$ (3.5W–V), and $0.2 \times 10^{-9}$ mol$_{NO}$ m$^{-2}$ s$^{-1}$ (V–W/TiO$_2$) in a dry condition (Fig. 5a and Supplementary Table 3), indicating that the reaction site of 3.5W–V was favorable for NH$_3$-SCR to proceed compared to the reaction sites of 0W–V and V–W/TiO$_2$. The reaction rate of 0W–V considerably decreased to $1.4 \times 10^{-9}$ mol$_{NO}$ m$^{-2}$ s$^{-1}$ in a wet condition and that of 3.5W–V was $4.2 \times 10^{-9}$ mol$_{NO}$ m$^{-2}$ s$^{-1}$. Thus, the inhibitory effect of water on the reaction was less for the tungsten-substituted vanadium oxide catalyst.

The apparent activation energy ($E_a$) values calculated from Arrhenius plots were 39 kJ mol$^{-1}$ (0W–V) and 36 kJ mol$^{-1}$ (3.5W–V) under a dry condition (Fig. 5b and Supplementary

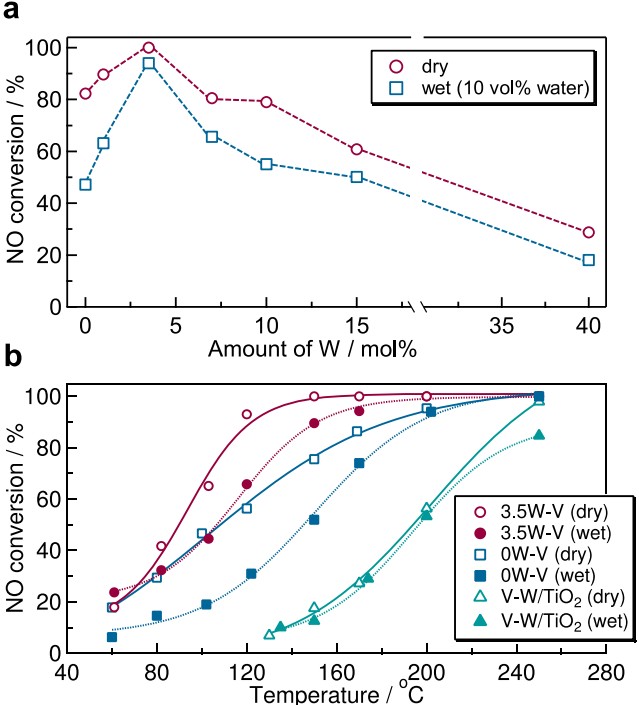

**Fig. 3 NH$_3$-SCR activity of W-substituted vanadium oxide catalysts. a** NO conversion (150 °C) of W-substituted vanadium oxide catalysts as a function of the molar ratio of tungsten. **b** NO conversion of 3.5W–V (W-substituted vanadium oxide), 0W–V (without tungsten), and V–W/TiO$_2$ (model of a conventional catalyst) as a function of reaction temperature. Reaction conditions: the amount of the catalyst, 0.375 g; reaction gas mixture, 250 ppm NO, 250 ppm NH$_3$, 4 vol% O$_2$ and 10 vol% H$_2$O (when used) in Ar; flow rate, 250 mL min$^{-1}$; space velocity, 40,000 mL h$^{-1}$ g$_{cat}$$^{-1}$.

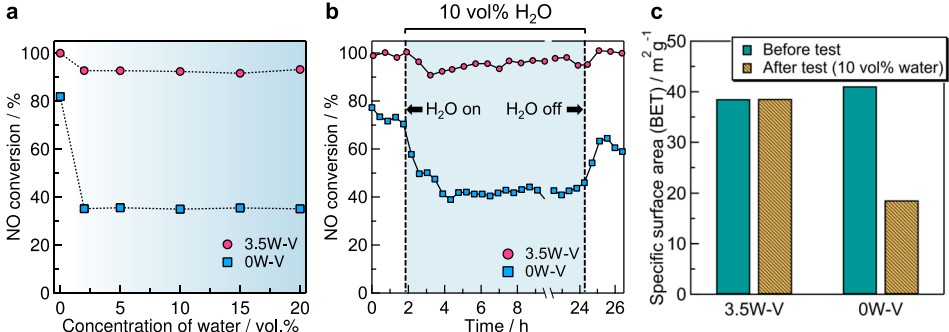

**Fig. 4 Water tolerance of W-substituted vanadium oxide catalysts. a** NO conversion (150 °C) of 3.5W–V (W-substituted vanadium oxide) and 0W–V (without tungsten) as a function of the concentration of water. **b** Catalytic stability test (150 °C) of 3.5W–V and 0W–V. Reaction conditions: the amount of the catalyst, 0.375 g; reaction gas mixture, 250 ppm NO, 250 ppm $NH_3$, 4 vol% $O_2$ and 2–20 vol% $H_2O$ (when used) in Ar; flow rate, 250 mL min$^{-1}$; space velocity, 40,000 mL h$^{-1}$ g$_{cat}^{-1}$. **c** Specific surface areas of 3.5W–V and 0W–V before and after a stability test.

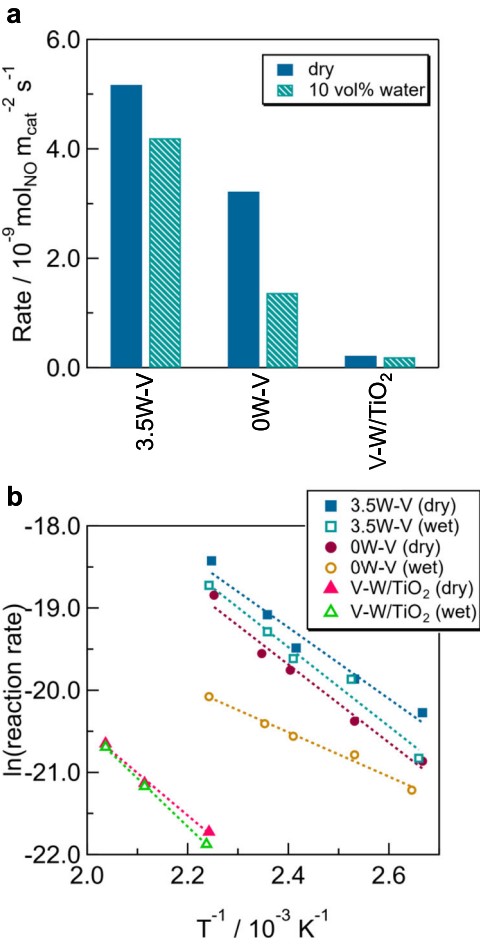

**Fig. 5 Kinetic measurements for $NH_3$-SCR. a** Reaction rate per specific surface area at 150 °C and **b** Arrhenius plots for 3.5W–V (W-substituted vanadium oxide), 0W–V (without tungsten), and V–W/TiO$_2$ (model of a conventional catalyst) at 150 °C under a dry condition and wet condition (10 vol% water).

Table 3), indicating that the reaction mechanisms were similar under a dry condition. $E_a$ of 3.5W–V under a wet condition was 40 kJ mol$^{-1}$, which was close to that under a dry condition, but that of 0W–V was 22 kJ mol$^{-1}$ and preexponential factor (intercept of the line) decreased in the presence of water vapor. The change in $E_a$ of 0W–V reflects the different reaction mechanism of vanadium oxide without tungsten. Although the $E_a$ of 0W–V became smaller, the number of active sites

(preexponential factor) would decrease because of its blockage by water, leading to a decrease in the activity. $E_a$ values of V–W/TiO$_2$ were 43 (dry) and 49 kJ mol$^{-1}$ (wet). Therefore, the effect of tungsten on the $NH_3$-SCR cycle would be similar to that in the bulk V–W oxide system.

The $NH_3$-SCR of a vanadium-based catalyst proceeds as follows: (1) adsorption of $NH_3$ on an acid site, (2) reaction of NO with $NH_3$ to produce $N_2 + H_2O$ via a nitrosamide (NH$_2$NO*) intermediate with simultaneous reduction of the vanadia surface, and (3) re-oxidation of the partially reduced vanadia surface by oxygen molecules[1,52,53]. Therefore, we measured the reaction orders for $NH_3$, NO, and $O_2$ at 150 °C (Supplementary Fig. 9 and Supplementary Table 3). Ranges of gas concentration were 125–500 ppm for $NH_3$, 125–500 ppm for NO, and 2–8% for $O_2$. Reaction orders of 0W–V and 3.5W–V were 0.2–0.3 ($NH_3$), 0.9–1.1 (NO), and 0.3–0.4 ($O_2$) under both dry and wet atmospheres. The small reaction order for $NH_3$ indicates that the adsorption of $NH_3$ to the catalyst surface is strong. The reaction order for NO was high compared to those for the other substrates, indicating that the reaction rate depended on NO concentration in actual conditions. The reaction order for $O_2$ was 0.3–0.4 at 150 °C, although zero-order dependence was reported at $O_2$ concentrations >1% at a higher temperature (>250 °C) for a supported vanadia catalyst[2,38,53]. Oxygen molecules contribute to the re-oxidation step of the partially reduced vanadia surface (V$^{4+}$ → V$^{5+}$) in the reaction mechanism of $NH_3$-SCR. However, the re-oxidation rate can be slower for a low temperature and the surface of the catalyst is not fully oxidized in a low-temperature range[2,38,53,54].

**Roles of tungsten substitution in the $NH_3$-SCR cycle.** We carried out *operando* IR and UV–Vis measurements at 150 °C to examine the reaction mechanism of $NH_3$-SCR over tungsten-substituted vanadium oxide catalysts at a low temperature. To observe the behavior of acid sites, we conducted *operando* IR measurements for 0W–V (V$_2$O$_5$ without tungsten), 3.5W–V (the most active catalyst), and 40W–V (with excess tungsten) as shown in Fig. 6. We flowed 0.1% $NH_3$/He to the IR discs of the catalysts to adsorb ammonia and purged excess ammonia with He. Then, the time course of the IR spectra was monitored under the condition of 500 ppm NO + 8% $O_2$/He flow with simultaneous detection of $N_2$ by a mass spectrometer (MS) equipped at the outlet of the IR cell. During this measurement, adsorbed $NH_3$ reacts with NO followed by reduction of redox-active sites. Then, reduced sites are re-oxidized by oxygen. The reaction would continue until adsorbed $NH_3$ is completely consumed (Supplementary Fig. 10). The amount of ammonia species adsorbed on Lewis and Brønsted acid sites was calculated from the area of corresponding IR peaks. Under the dry condition, IR peaks

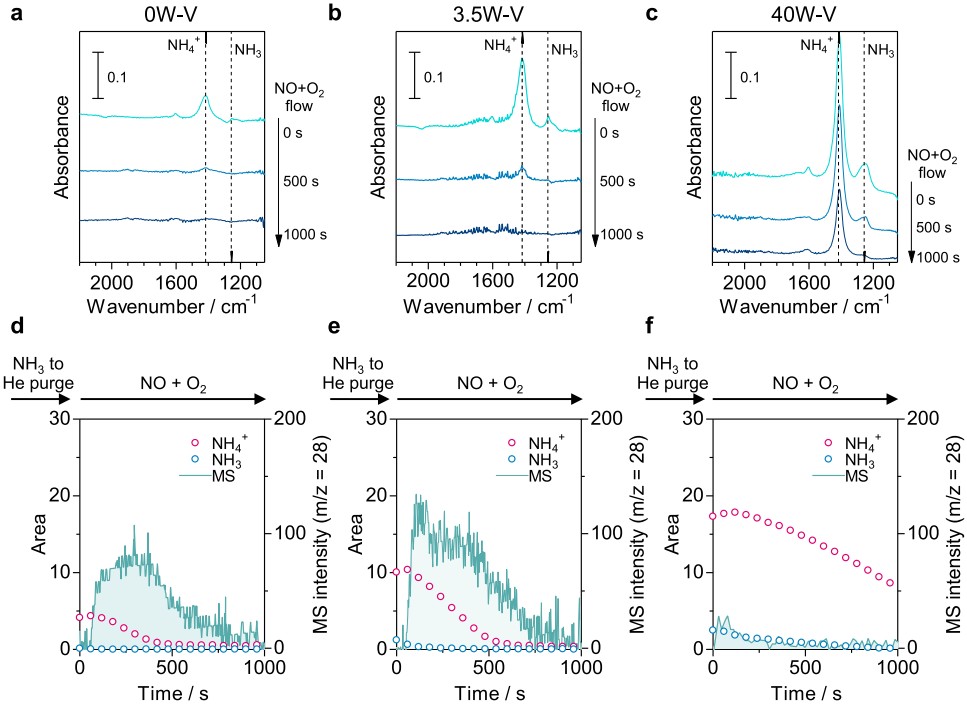

**Fig. 6 Acid properties under a dry atmosphere. a–c** *Operando* IR spectra of ad species on 0W–V (without tungsten), 3.5W–V (with tungsten), and 40W–V (excess tungsten) during NH$_3$ adsorption followed by NO + O$_2$ exposure at 150 °C under a dry condition. The IR disc was exposed to 0.1% NH$_3$/He flow (30 min) and purged with He (20 min), followed by exposure to 500 ppm NO + 8% O$_2$ (He balance) flow. **d–f** IR peak areas of ammonia species adsorbed on Lewis acid sites (NH$_3$) and Brønsted acid sites (NH$_4^+$) and MS intensity of N$_2$ versus time of NO + O$_2$ flowing.

assigned to NH$_3$ adsorbed on the Lewis acid site (1250–1254 cm$^{-1}$) and NH$_4^+$ adsorbed on Brønsted acid site (1410–1420 cm$^{-1}$) were observed for 0, 3.5, and 40W–V after the adsorption of ammonia (Fig. 6a–c). The relative amounts of Lewis and Brønsted acid sites, which were calculated from the area of initial IR peaks after ammonia adsorption (spectra at 0 s in Fig. 6a–c), were summarized in Fig. 7a. The numbers of acid sites increased from 0.1 to 2.5 (Lewis acid, NH$_3$) and from 4.1 to 17.3 (Brønsted acid, NH$_4^+$) with an increase in the molar ratio of tungsten in the catalysts up to 40 mol%. The results indicate that both Lewis and Brønsted acid sites are created by the incorporation of tungsten. The areas of the IR peaks for NH$_3$ and NH$_4^+$ decreased with time of NO + O$_2$ flow for 1000 s (Fig. 6a–c and d–f, red and blue lines). The consumption of ammonia species was completed within 500 s for 0 and 3.5W–V, while 40W–V showed a slow consumption rate. We also confirmed N$_2$ production upon the introduction of NO + O$_2$, and a greater amount of N$_2$ was observed for 3.5W–V than for 0W–V (Fig. 6d, e, mass spectra), indicating that the production of N$_2$ was facilitated by the addition of tungsten. On the other hand, the MS intensity of N$_2$ was considerably low for 40W–V (Fig. 6f, mass spectra). These results suggested that NH$_3$-SCR activity decreases when an excess amount of tungsten is introduced to vanadium oxide because of high tungsten coverage that leads to blocking of the surface vanadium species as catalytically active sites despite the increase in acid sites.

To investigate the redox properties of the catalysts, we also conducted *operando* UV–Vis measurements (Fig. 7b and Supplementary Figs. 11 and 12). We monitored the change in the pseudo-absorbance (Kubelka–Munk unit) of UV–Vis spectra at $\lambda = 700$ nm ($\Delta KM_{700}$), which is assigned to the $d$–$d$ transition of V$^{4+}$, to observe the change in the valence of vanadium sites (V$^{5+} \leftrightarrow$ V$^{4+}$). The production of N$_2$ was recorded simultaneously by an MS. The monitoring was performed under NO + NH$_3$ flowing followed by O$_2$ flowing to evaluate the respective reduction and oxidation half-cycles. The correlations between

W content, amount of V$^{5+}$ reduced by NO + NH$_3$ ($\Delta KM_{700}$ under NO + NH$_3$ flowing), and MS intensity for N$_2$ are shown in Fig. 7b. The 3.5W–V showed the largest $\Delta KM_{700}$ and MS intensity of N$_2$ and had superior reduction capability for the production of N$_2$. On the other hand, $\Delta KM_{700}$ considerably decreased when 40 mol% of tungsten was doped, indicating that the catalyst with an excess amount of tungsten was difficult to be reduced. The 3.5W–V also showed the largest $\Delta KM_{700}$ under O$_2$ flowing (Supplementary Figs. 11b, d, f and 12b). Thus, 3.5W–V with moderate tungsten loading turned out to be the most redox-active catalyst, leading to the largest N$_2$ production. Combining with the results of *operando* IR measurements, tungsten sites serve as acid sites for the adsorption of ammonia, which is stronger than those pre-existing in vanadium oxide without tungsten, and the vanadium (V$^{5+}$) sites work as redox sites for NH$_3$-SCR. The highest activity for 3.5W–V can be ascribed to the increase in the number of acid sites and the high redox capability of the surface V species. W–O–V units would be needed for the progression of the reaction, and the number of adjacent tungsten sites such as W–O–W units becomes dominant with an increase in the amount of tungsten, leading to blocking of the surface redox-active V sites and thus resulting in a decrease in activity.

**Effects of water on the NH$_3$-SCR cycle**. The activity test clearly showed a positive effect of tungsten substitution in the presence of water. We also conducted *operando* IR measurements under a wet condition (2 vol% water) to check the effect of water on the acid sites for 0W–V and 3.5W–V. Although the IR peaks of adsorbed ammonia species on Lewis and Brønsted acid sites (NH$_3$ and NH$_4^+$, respectively) were both observed under a dry condition, the absorption band attributed to the NH$_4^+$ species was exclusively seen in the presence of water (Fig. 8a, b). The areas of the IR absorption band for Lewis and Brønsted acid sites under dry and wet conditions are shown in Fig. 8e, f. The area of

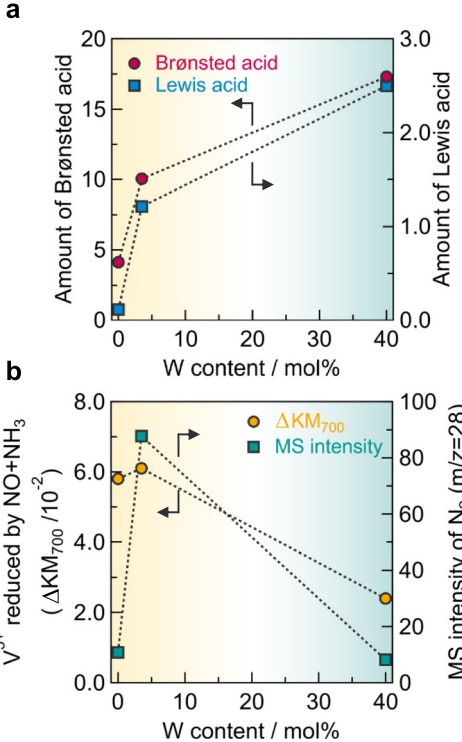

**Fig. 7 Effect of tungsten on acid and redox properties. a** Amount of Lewis and Brønsted acid sites as a function of W content. The amount of both acid sites was observed from the area of initial IR spectra after ammonia adsorption (spectra at 0 s in Fig. 6a–c). **b** Amount of $V^{5+}$ reduced by NO + $NH_3$ and MS intensity of $N_2$ produced as a function of W content. The amount of $V^{5+}$ reduced was determined from the change in the Kubelka–Munk unit at $\lambda = 700$ nm ($\Delta KM_{700}$) under NO (500 ppm) + $NH_3$ (500 ppm) flowing (1500 s) at 150 °C under a dry atmosphere shown in Supplementary Figs. 7 and 8a.

the IR absorption band for $NH_4^+$ (Brønsted acid sites) increased from 10.0 (dry) to 18.7 (wet) and that for $NH_3$ (Lewis acid sites) decreased from 1.21 (dry) to 0.27 (wet) in the case of 3.5W–V when water vapor was introduced (3.5W–V; Fig. 8d–f). On the other hand, 0W–V showed only a slight change in the distribution of acid sites (0W–V; Fig. 8c, e, f). On a metal oxide surface, Lewis acid sites (coordinatively unsaturated metal cations) and Brønsted base sites (oxygen ions) can be changed into Brønsted acid sites by dissociative adsorption of water[58]. In a previous study, it was theoretically predicted that Brønsted acid sites (hydroxyl groups) are newly created on the $V_2O_5$–$WO_3$ solid solution by dissociative adsorption of water on tungsten-exposed sites and bridging oxygen sites under a wet atmosphere (Fig. 8g)[48]. On the other hand, water could be adsorbed on $V_2O_5$ without tungsten, but a Brønsted acid site does not form[47,48]. The observed changes in the surface state of vanadium oxide with and without tungsten can reflect such theoretical predictions. The area of the IR spectra ($NH_4^+$) decreased with time of NO + $O_2$ flowing both for 3.5W–V and 0W–V (Fig. 8c, d, red and blue lines), and ammonia species were consumed by the reaction under a wet condition. Considerable $N_2$ production was confirmed for 3.5W–V under a wet condition, but the $N_2$ production of 0W–V was smaller than that under a dry condition (Fig. 8c, d, mass spectra). These results suggest that $NH_3$-SCR of bulk vanadium oxide without tungsten is strongly inhibited by water, while bulk tungsten-substituted vanadium oxide proceeds with

the reaction by newly created protonic Brønsted acid sites that are less affected by water.

*Operando* UV–Vis measurements were also conducted for 3.5W–V with and without water vapor to check the redox cycle (Fig. 9 and Supplementary Fig. 13). NO (500 ppm) + $NH_3$ (500 ppm) and $O_2$ (8%) were repetitively introduced three times to confirm the redox cycle. During the measurements under wet and dry conditions, the pseudo-absorbance (KM unit) at $\lambda = 700$ nm similarly increased and decreased under NO + $NH_3$ and $O_2$ flowing, showing reduction ($V^{5+} \rightarrow V^{4+}$) and oxidation ($V^{4+} \rightarrow V^{5+}$) half-cycles (Fig. 9a–c for a wet condition and Supplementary Fig. 13 for a dry condition). The times of change in pseudo-absorbance (KM unit) at $\lambda = 700$ nm are shown by blue (wet) and red (dry) solid lines in Fig. 9d. Repetitive production of $N_2$ with a change in $\Delta KM_{700}$ was confirmed under both dry and wet atmospheres (Fig. 9d). Notably, considerable $N_2$ production was seen even in the presence of water (Fig. 9d, mass spectra). These results demonstrate that 3.5W–V proceeds with $NH_3$-SCR by the redox cycle and the $N_2$ production is not affected by water in a transient state.

**Difference between supported and bulk vanadium oxide.** *Operando* IR spectra were observed for 3.5W–V and V–W/TiO$_2$ to consider the difference between bulk and supported tungsten-substituted vanadium oxide catalysts. The measurement was conducted at 200 °C at which V–W/TiO$_2$ is sufficiently active and NO was flowed into samples after adsorption of $NH_3$ to obtain the information of acid sites next to redox sites (Supplementary Fig. 14). After the adsorption of $NH_3$ under a wet condition, $NH_4^+$ species adsorbed on Brønsted acid sites were solely confirmed for 3.5W–V (Fig. 10a), which was a similar result obtained at 150 °C in the presence of water. $NH_4^+$ species were rapidly consumed and significant $N_2$ production was confirmed when NO flowed (Fig. 10c). On the other hand, $NH_3$ species adsorbed on Lewis acid sites were mainly observed in addition to $NH_4^+$ species for V–W/TiO$_2$ (Fig. 10b). However, the consumption of $NH_4^+$ and $NH_3$ species were dull and $N_2$ production was less (Fig. 10c). The results demonstrate that Brønsted acid sites of 3.5W–V are more reactive than Brønsted and Lewis acid sites of V–W/TiO$_2$. After the above *operando* IR measurement under NO flowing for 3.5W–V, the sample was oxidized and NO flowed again to check if the residual $NH_4^+$ reacts with NO. Although the $N_2$ production and the consumption of $NH_4^+$ were attenuated under the first NO flowing (Fig. 10c), $NH_4^+$ species on Brønsted acid sites were consumed again and $N_2$ production was confirmed under the second NO flowing after oxidation (Supplementary Fig. 15). If *Operando* UV–Vis measurement was conducted in the same procedure ($NH_3 \rightarrow$ 1st NO $\rightarrow O_2 \rightarrow$ 2nd NO), the similar $N_2$ production behavior was observed accompanied by redox cycle ($V^{5+} \leftrightarrow V^{4+}$) under the first and second NO flowing (Supplementary Fig. 16). These results show that V sites reduced by the reaction of $NH_4^+$ (Brønsted acid site) with NO are re-oxidized, and remaining ammonia species move to Brønsted acid sites which are adjacent to redox V sites, then they react with NO by the re-oxidized V sites again. According to the previous reports on supported vanadium-based catalysts, Brønsted acid sites are not directly involved in the catalytic cycle and they play a role in an $NH_3$ pool to supply $NH_3$ to the Lewis acid sites[20,52]. In the case of bulk tungsten-substituted vanadium oxide catalyst, Brønsted acid site (B) would be located next to redox-active $V^{5+}$ site and it would directly react with NO as following reduction half-cycle:

$$V(V) = O + NH_3(B) + NO \rightarrow V(IV) - OH + N_2 + H_2O.$$

$$(2)$$

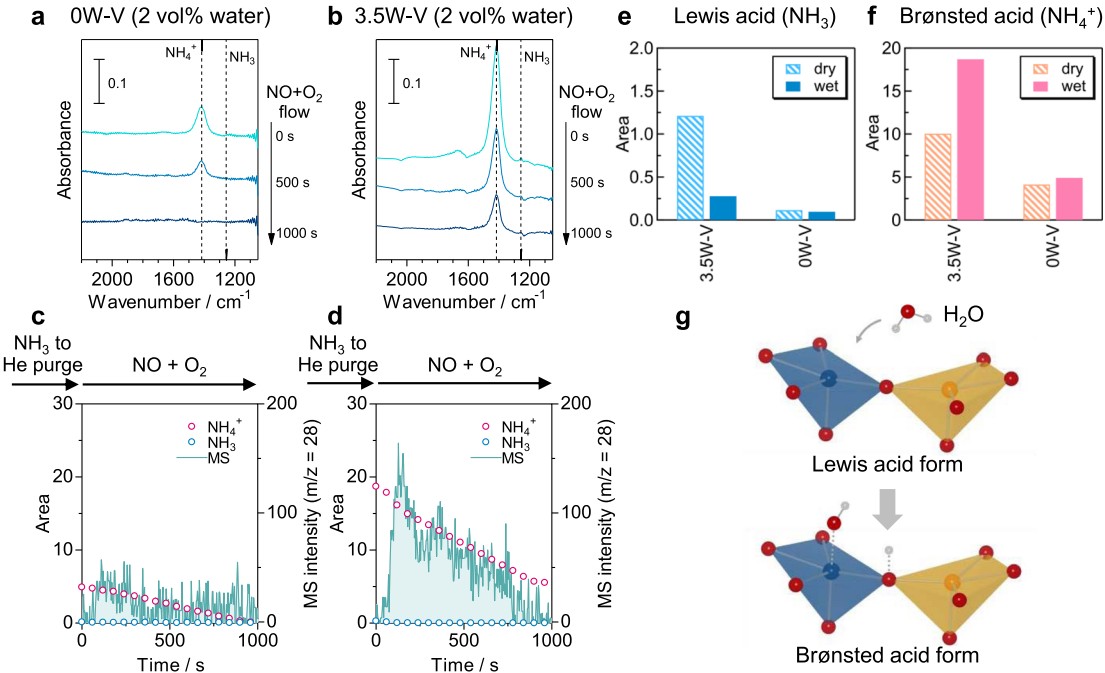

**Fig. 8 Acid properties in the presence of water vapor. a, b** *Operando* IR spectra of ad species on 0W–V (without tungsten) and 3.5W–V (with tungsten) during $NH_3$ adsorption, followed by $NO + O_2$ exposure at 150 °C under a wet condition (2 vol% water). The IR disc was exposed to 0.1% $NH_3$/He flow (30 min) and purged with He (20 min), followed by exposure to 500 ppm $NO + 8\% O_2 + 2\% H_2O$ (He balance) flow. **c, d** The IR peak areas of $NH_3$ adsorbed on Lewis acid sites ($NH_3$) and Brønsted acid sites ($NH_4^+$) and MS intensity of $N_2$ versus time of $NO + O_2$ flowing. IR areas for **e** Lewis acid and **f** Brønsted acid measured from IR spectra (0 s) under dry and wet conditions. **g** Metal-exposed surface of W-substituted vanadium oxide and its changes from Lewis acid form to Brønsted acid form. Orange polyhedra: vanadium units; blue polyhedra: tungsten units.

Furthermore, we conducted temperature-programmed reaction measurements under $NO + NH_3$ flow for 3.5W–V and V–W/$TiO_2$ (Supplementary Fig. 17) to observe the reducibility. The reduction of 3.5W–V started at ambient temperature, while a higher temperature was needed to reduce V–W/$TiO_2$. Bulk tungsten-substituted vanadium oxide catalyst has water-tolerant Brønsted acid sites and greater reducibility originated from bulk characteristics, resulting in high $NH_3$-SCR activity at a low temperature in the presence of water vapor.

## Discussion

We investigated the low-temperature $NH_3$-SCR activity of tungsten-substituted vanadium oxide and the reaction mechanism under dry and wet conditions. The catalysts, 0–40 mol% tungsten-substituted vanadium oxide, were synthesized from ammonium metavanadate ($NH_4VO_3$) and ammonium metatungstate by the oxalate method. We confirmed from atomic-resolution HAADF-STEM images that the lattice vanadium sites were substituted by tungsten atoms, while adjacent and cluster tungsten moieties were found when an excess amount of tungsten was doped up to 40 mol%. XRD measurements also showed that the $WO_3$ phase was generated in catalysts with an increase in the molar ratio of >3.5 mol%. The 3.5W–V catalyst showed the highest NO conversions, >99% (dry) and ~93% (wet, 10 vol% water). The NO conversion decreased when >3.5 mol% of tungsten was added to vanadium oxide. The stability of the catalysts was increased by the addition of tungsten because bulk $WO_6$ units connected vanadium oxide layers. The reaction mechanism was investigated for 0W–V ($V_2O_5$ without tungsten), 3.5W–V (the best active catalyst), and 40W–V (excess tungsten) using *operando* IR and UV–Vis measurements. It was found that the vanadium site plays a role as a redox site and the tungsten site

contributed to the acid sites for adsorption of ammonia. With an increase in tungsten, the activity decreased because the number of redox sites decreased for high tungsten coverage despite the increase in the number of acid sites. Acid sites of tungsten-substituted vanadium oxide were converted to Brønsted acid sites under a wet condition, while vanadium oxide without tungsten did not show a significant change in the population. From *operando* UV–Vis measurements, 3.5 mol% tungsten-substituted vanadium oxide showed high redox capabilities even in the presence of water, and $N_2$ production during the reduction half-cycle was hardly affected by the addition of water. The 3.5W–V catalyst had the high redox ability and reactivity of Brønsted acid sites at a low temperature in the presence of water. These results evidence water tolerance of bulk tungsten-substituted vanadium oxide. Although our results experimentally demonstrated the effect of water on the $NH_3$-SCR cycle over the V–W oxide system, the detailed reaction mechanism needs to be elucidated since water molecules may participate in each elementary reaction step. The acid and redox properties and the effect of water would also change depending on various factors such as the morphology, exposed facet, and surface reconstruction. We believe that our results provide a better understanding of $NH_3$-SCR at a low temperature, which is a future task for the current deNO$x$ process.

## Methods

**Reagents**. $NH_4VO_3$ and oxalic acid were purchased from FUJIFILM Wako Pure Chemical Corporation. Ammonium metatungstate (($NH_4$)$_6$[$H_2W_{12}O_{40}$]·$nH_2O$, $n ≒ 6$) was purchased from Nippon Inorganic Colour & Chemical Co., Ltd. All reagents were used without further purification.

**Synthesis of tungsten-substituted vanadium oxide catalysts**. First, $NH_4VO_3$ and oxalic acid (11.9 g, 131.7 mmol) were dissolved in 50 mL of water and the aqueous solution was stirred for 10 min for completion of the change in color of

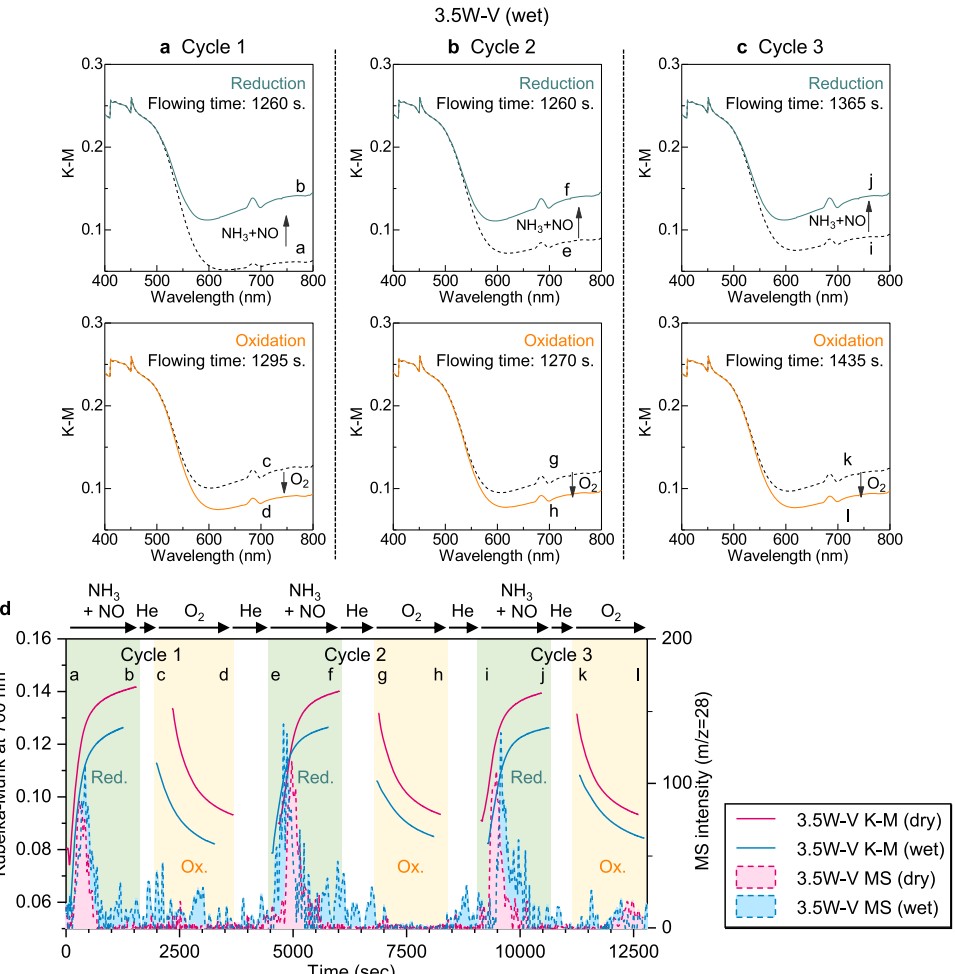

**Fig. 9 Redox properties of W-substituted vanadium oxide (3.5W–V) under wet and dry conditions.** *Operando* UV–Vis spectra of 3.5W–V for reduction (500 ppm NO + 500 ppm NH₃) and oxidation (8% O₂) half-cycles at 150 °C under a wet (2 vol% water) atmosphere for the **a** first cycle, **b** second cycle, and **c** third cycle. **d** Change in the Kubelka–Munk unit at λ = 700 nm and MS intensity of N₂ as a function of time for 3.5W–V during NO (500 ppm) + NH₃ (500 ppm) and O₂ (8%) flowing at 150 °C under a dry atmosphere and wet (2 vol% water) atmosphere. The characters a–l correspond to the points at which the UV–Vis spectra were observed.

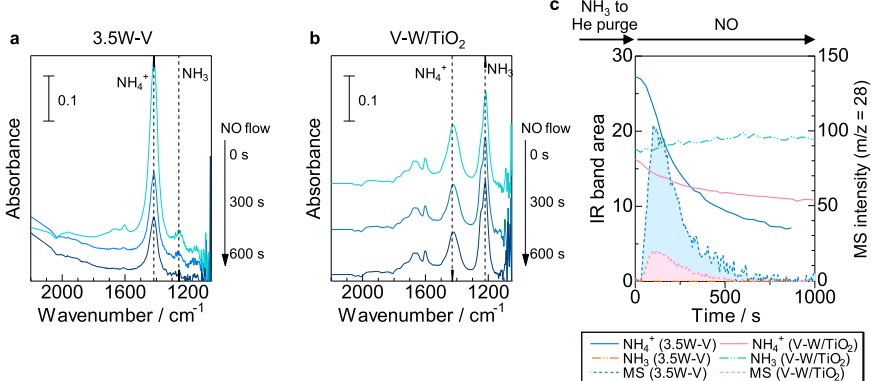

**Fig. 10 Difference between bulk and supported catalysts.** *Operando* IR spectra of ad species on **a** 3.5W–V (bulk W-substituted vanadium oxide) and **b** V–W/TiO₂ (supported catalyst) during NH₃ adsorption in the presence of water followed by NO exposure at 200 °C. The IR disc was exposed to 0.1% NH₃/He flow in the presence of 2% water vapor (30 min) and purged with He (20 min), followed by exposure to 500 p.pm NO (He balance) flow. **c** IR peak areas of ammonia species adsorbed on Brønsted acid sites (NH₄⁺) and MS intensity of N₂ versus time of NO flowing.

the solution. Next, ammonium metatungstate was added to the solution. Then, the solution was heated overnight to evaporate the water on a hotplate. Finally, the resulting solid was calcined twice at 300 °C for 4 h each time. The sample was denoted as $x$W–V, where $x$ is the molar ratio (mol%) of W to V. The amounts of reagents are shown in Supplementary Table 1.

**Synthesis of TiO₂-supported V₂O₅/WO₃ (1 wt% V₂O₅–5 wt% WO₃/TiO₂).** First, vanadium and tungsten precursor aqueous solutions were prepared. For the vanadium precursor aqueous solution, 0.020 g of NH₄VO₃ (0.17 mmol) and 0.045 g of oxalic acid (0.51 mmol) were dissolved in 2 mL of water and the aqueous solution was stirred for 10 min. For the tungsten precursor aqueous solution, 0.084

g of ammonium metatungstate hydrate (0.028 mmol) and 0.089 g of oxalic acid (0.99 mmol) were dissolved in 2 mL of water and the aqueous solution was stirred for 10 min. Then, 1.4 g of $TiO_2$ (P25) was added to 20 mL water and the vanadium and tungsten precursor aqueous solutions were added to the dispersion. The dispersion was then heated at 120 °C to evaporate the water. Finally, the resulting solid was calcined at 300 °C for 4 h.

**Catalyst characterization**. STEM images were obtained by using FEI Titan Cubed G2 60–300. For TEM measurements, samples were deposited on a carbon film-coated mesh copper grid. XRD patterns were collected by SmartLab (Rigaku) with Cu Kα radiation. Rietveld analyses were carried out using Reflex program in Materials Studio 2017.

**Catalytic activity test**. The $NH_3$-SCR activity of vanadium oxide catalysts was measured using a fixed-bed flow reactor (Supplementary Fig. 1). The reaction gas mixture, 250 ppm NO, 250 ppm $NH_3$, 4 vol% $O_2$ and 5–20 vol% water (when used) in Ar (250 mL min$^{-1}$), was fed to the catalyst (0.375 g). The outlet gases were analyzed by an IR spectrometer (JASCO FT/IR-4700) equipped with a gas cell (JASCO LPC12M-S). NO conversion and $N_2$ selectivity were calculated by the following equations:

$$\mathrm{NO\ conversion}(\%) = \frac{NO_{in} - NO_{out}}{NO_{in}} \times 100, \quad (3)$$

$$N_2\ \mathrm{selectivity}(\%)\ \frac{2 \times N_{2out}}{(NO_{in} + NH_{3in}) - (NO_{out} + NH_{3out})}$$
$$(2 \times N_{2out} = (NO_{in} + NH_{3in}) - (NO_{out} + NH_{3out} + NO_{2out} + 2 \times N_2O_{out})). \quad (4)$$

Reaction rate and reaction order of the catalysts were determined by adjusting the amount of catalysts and the flow rate for NO conversion to be below 20%.

**Operando FT-IR measurements**. *Operando* FT-IR spectra were recorded at 150 °C using a JASCO FT/IR-4200 with a TGS (triglycine sulfate) detector. Samples (each 40 mg) were pressed to obtain self-supporting pellets ($\phi = 20$ mm), which were placed in a quartz IR cell with $CaF_2$ windows connected to a conventional gas flow system. Prior to measurements, the sample pellets were heated under a flow of 10% $O_2$/He (100 mL min$^{-1}$) at 300 °C for 10 min. $NH_3$/He (0.1%) was then introduced for 0.5 h, and subsequently, the gas was switched to He (20 min) in order to let the residual $NH_3$ gas out. After taking the first spectrum, 500 ppm NO, 8% $O_2$ (when used)/He was introduced to the sample at a flow rate of 100 mL min$^{-1}$. For the measurement under a wet condition, 2% $H_2O$ was introduced. Spectra were measured by accumulating 20 scans at a resolution of 4 cm$^{-1}$. A reference spectrum taken at 150 °C under He flow was subtracted from each spectrum. An MS (BELMass, MicrotracBEL Corp.) was used for the analysis of $N_2$ gas.

**Operando UV-Vis measurements**. Diffuse reflectance UV–Vis measurements were conducted at 150 °C with a UV–Vis spectrometer (JASCO V-670) connected to an *operando* flow cell with a quartz window. The light source was led to an integrating sphere through an optical fiber. Samples (each 10 mg) were placed in the sample cell connected to a gas flow system. Reflectance was converted to KM units using the KM function. A background spectrum was corrected by measuring $BaSO_4$. Prior to measurements, the sample pellets were heated under a flow of 10% $O_2$/He (100 mL min$^{-1}$) at 300 °C for 10 min. Then, the sample pellets were cooled down to 150 °C. The measurements were performed by the following steps: (i) The sample pellets were purged with He flow to remove residual oxygen and the initial UV–Vis spectra were measured (spectra 1). (ii) A gas mixture (500 ppm $NH_3$ + 500 ppm NO/He) was fed to a sample at a flow rate of 100 mL min$^{-1}$ for a given time. Then, UV–vis spectra were recorded (spectra 2). (iii) After turning off the gas flow, the sample pellets were purged with He flow followed by UV–Vis spectra acquisition (spectra 3). (iv) Then, a gas mixture (8% $O_2$/He) was fed to a sample at a flow rate of 100 mL min$^{-1}$ for a given time and UV–Vis spectra were obtained (spectra 4). Steps (i)–(iv) were regarded as one cycle. The change in the KM unit at $\lambda = 700$ nm ($\Delta KM_{700}$), which was assigned to the $d$–$d$ transition of $V^{4+}$, was measured to obtain the relative amount of redox sites. $\Delta KM_{700}$ values for reduction ($V^{5+}$ reduced by NO + $NH_3$) and oxidation ($V^{4+}$ oxidized by $O_2$) half-cycles were calculated by the following equations:

$$(\Delta KM_{700}\ \text{for reduction half cycle}) = (KM_{700}\ \text{in spectra 2}) - (KM_{700}\ \text{in spectra 1}), \quad (5)$$

$$(\Delta KM_{700}\ \text{for oxidation half cycle}) = (KM_{700}\ \text{in spectra 4}) - (KM_{700}\ \text{in spectra 3}). \quad (6)$$

The production of $N_2$ in the outlet gas mixture was monitored by an MS (BELMass, MicrotracBEL Corp.).

**Others**. The crystal structure was drawn by VESTA[59].

## Data availability

The data that support the findings of this study are available from the corresponding author upon reasonable request.

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

## Acknowledgements

This work was supported, in part, by the Cooperative Research Program of Institute for Catalysis, Hokkaido University (20B1021), Nanotechnology Platform Program of the Ministry of Education, Culture, Sports, Science and Technology (MEXT), Japan and JSPS KAKENHI (20K15092).

## Author contributions

Y.I. and T.M. conceived the original concept and designed the experiments. Y.I. conducted the experiments and wrote the manuscript. H.K. carried out the operando UV measurements and helped Y.I. to measure operando IR spectra. T.T. and K.-i.S. supervised the operando measurements and supported the analysis of the results. S.I. and W.U. conducted the Rietveld analysis. S.H. helped synthesis catalysts. N.S. carried out HAADF-STEM measurements. E.K., K.M., and K.Y. supported catalytic activity test. M.H. gave advice about the interpretation of the results. T.M. supervised this project. All authors contributed to discuss the results.

## Competing interests
The authors declare no competing interests.
