## [Peer Review File · Nature Communications]

REVIEWER COMMENTS

Reviewer #1 (Remarks to the Author):

The manuscript deals with an interesting investigation of W doped bulk V₂O₅ for the low temperature NH₃-SCR reaction, with particular focus on the effect of water on the activity, stability, acid and redox properties of the surface sites

I do not have specific issues to be addressed, only a small doubt about the last sentence of page 10 (line 135), which seems to include a repetition:

'indicating that cluster formation or phase separation of tungsten oxide partially occurs with an increase in the amount of tungsten for a high tungsten content'

Reviewer #2 (Remarks to the Author):

The review summary for “Low-temperature selective catalytic reduction (NH₃-SCR) in the presence of water: Effect of tungsten substitution on a bulk vanadium oxide catalyst”

The manuscript introduced a W-substituted vanadium oxide catalysts for NH₃-SCR at a low temperature in the presence of water (~20 vol.%). Operando spectroscopy was used to provide insight into reaction mechanism. The manuscript was well written. However, the novelty of the manuscript was not justified. This manuscript is delivering the concept that in the presence of water, the W-V oxide could be tolerant due to the increased Bronsted acid sites arising from the dissociative adsorption of water while pure V oxide is not able to achieve the same objective. Unfortunately, this idea has already been demonstrated by other groups that the dissociative adsorption of water occurs on adjacent V-O-W sites, leading to increase of Bronsted acid sites. Meanwhile, the water adsorption behavior has also been extensively studied on V-W oxide system by other groups. Because of this, I am very sorry that I could not recommend it to be published in such a top journal because of the novelty issue. The followings are my comments for the authors.

1. What are the different roles of Bronsted and Lewis acid sites in the NH₃-SCR in terms of reaction rate and pathway?
2. It was stated in the manuscript text that “The role of water is thought to inhibit the adsorption by influencing adsorption of the reactant and/or the reaction between NO and adsorbed NH₃”. Either strong experimental evidence or references should be provided to support this mechanism on the effect of water in SCR.
3. What are crystal structure and morphology of samples loaded with different W content after reaction in the presence of water?
4. In the presence of water, water is dissociatively adsorbed on the W-V oxide. The hydroxyl groups are serving as Bronsted acid site. The remaining question on the role of proton or hydride must be

addressed.

Reviewer #3 (Remarks to the Author):

In this manuscript, the low-temperature NH₃-SCR reaction mechanism over W-substituted vanadium oxide catalysts in the presence of water was investigated by various experimental characterizations. The authors found that the 3.5 mol% W-substituted vanadium oxide showed excellent NH₃-SCR performance at low temperature, which was attributed to the fact that the moderate tungsten loading facilitated the formation of Brønsted acid sites in the presence of water, and maintained high redox ability of the vanadium oxide in the meanwhile. The finding is interesting and is important for the development of SCR catalyst, however, the quality of the manuscript should be substantially improved before the reconsideration. Further clarification and correction for the issues listed below is required.

1. The SCR activity test was carried out under 10 vol.% water (Figure 3), while the operando IR and UV-Vis measurements (Figure 8 and 9) were performed under 2 vol.% water. Why didn't the authors use the same humidity conditions? What are the IR and UV-Vis results under 10 vol.% water?

2. Although the author claims that the prepared vanadium oxide catalyst with bulk crystal structure has low oxidation ability of SO₂, this point is doubtful. The author should show the SCR of NO_x activity with the presence of SO₂, and the conversion of SO₂.

3. The authors claim that the apparent activation energy (E_a) for the NH₃-SCR over the OW-V catalyst in the presence of water (22 kJ mol⁻¹) is lower than that without water (39 kJ mol⁻¹, Line 213 and Table 1). However, the NH₃-SCR activity in the presence of water is lower than that under dry condition (Figure 3). The two results are contradictory.

4. As shown in Figure 9d, the amounts of N₂ production under wet condition seems to be higher than that under dry condition, instead of "almost the same" (line 313 of page 26), which is also contradicts the activity results (Figure 3).

5. In some recent important progresses of SCR research (Angew. Chem. Int. Ed. 2016, 55, 11989; Sci. Adv. 2018, 4, eaau4637), Lewis acid sites have been demonstrated to be the major active sites, while Brønsted acid sites are not directly involved in the catalytic reaction and mainly serve as an NH₃ pool to replenish the Lewis acid sites. This study shows that the NH₃-SCR activity under dry condition is higher than that in the presence of water and the excess tungsten substitution decreases the activity (Figure 3), which are consistent with the above conclusion. However, the authors also emphasized that importance of Brønsted acid sites. These understandings need to be unified.

6. There are many spelling errors. For instance, in the caption of Figure 1, the second "(b)" and "(c)" should be "(d)" and "(e)", respectively, and the "Enralrged" should be "Enlarged". In the line 133 of page 10, the "osberbed" should be "observed". In the caption of Figure 2, the "Enralrged" should be "Enlarged", and the "010" should be corrected to "(010)". In Figure 7b, the "ΔKM300" should be corrected to "ΔKM700. In the caption of Figure 8, the "(e) Brønsted acid and (f) Lewis acid" should be corrected to "(e) Lewis acid and (f) Brønsted acid". In the line 143 and 288, the "figure" should be "Figure". Also, the language should be substantially improved with professional assistance.

Responses to Referees' Comments

The following changes were made in the new manuscript. The revisions follow the outline of the questions from each reviewer. We also made editorial corrections based on the Nature Communications formatting instruction.

Response to reviewer #1:

We appreciate reviewer 1 for careful and profound reading of this manuscript and the helpful comments. We have answered your points below.

Comments:

The manuscript deals with an interesting investigation of W doped bulk V₂O₅ for the low temperature NH₃-SCR reaction, with particular focus on the effect of water on the activity, stability, acid and redox properties of the surface sites

I do not have specific issues to be addressed, only a small doubt about the last sentence of page 10 (line 135), which seems to include a repetition:

'indicating that cluster formation or phase separation of tungsten oxide partially occurs with an increase in the amount of tungsten for a high tungsten content'

Response:

We would like to thank the reviewer for the positive feedback and thoughtful comments. The sentence that reviewer 1 pointed out was corrected to be a simpler expression as below:

- A tungsten oxide phase partially forms with an increase in the amount of tungsten.
(Page10, line133)

Response to reviewer #2:

We appreciate reviewer 2 for careful and profound reading of this manuscript and the helpful comments and useful suggestions for further improvement of this manuscript. We have answered your points below.

Comments:

The review summary for “Low-temperature selective catalytic reduction (NH₃-SCR) in the presence of water: Effect of tungsten substitution on a bulk vanadium oxide catalyst”

The manuscript introduced a W-substituted vanadium oxide catalysts for NH₃-SCR at a low temperature in the presence of water (~20 vol.%). Operando spectroscopy was used to provide insight into reaction mechanism. The manuscript was well written. However, the novelty of the manuscript was not justified. This manuscript is delivering the concept that in the presence of water, the W-V oxide could be tolerant due to the increased Brønsted acid sites arising from the dissociative adsorption of water while pure V oxide is not able to achieve the same objective. Unfortunately, this idea has already been demonstrated by other groups that the dissociative adsorption of water occurs on adjacent V-O-W sites, leading to increase of Brønsted acid sites. Meanwhile, the water adsorption behavior has also been extensively studied on V-W oxide system by other groups. Because of this, I am very sorry that I could not recommend it to be published in such a top journal because of the novelty issue. The followings are my comments for the authors.

Response: We would like to thank the reviewer for the feedback and thoughtful comments. As reviewer #2 pointed out, it has previously been reported that Brønsted acid sites can form on Lewis acid sites by dissociative adsorption of water. However, the novel concepts that we would like to suggest in this paper are:

1. The bulk W-substituted vanadium oxide has Brønsted acid sites in which the adsorption of NH₃ is not inhibited by water in the NH₃-SCR cycle.
2. NH₃ species adsorbed on Brønsted acid sites (NH₃(B)) react with NO accompanied by the reduction of V⁵⁺ sites even at a low temperature and proceed with reduction half cycle by their displacement to V⁵⁺ sites as the following equation:
$$V(V)=O + NH_3(B) + NO \rightarrow V(IV)-OH + N_2 + H_2O$$
3. The reactivity of NH₃ species adsorbed on Brønsted acid sites of the bulk W-substituted vanadium oxide is higher toward the reduction half cycle compared to those of Brønsted acid and Lewis acid sites of a conventional V₂O₅/WO₃/TiO₂ catalyst.

The above concepts and insights have not been provided although supported $V_2O_5/WO_3/TiO_2$ has been extensively investigated for NH_3 -SCR and recent works have emphasized the importance of Lewis acid sites. We conducted additional measurements for conventional $V_2O_5/WO_3/TiO_2$ and bulk $W-V_2O_5$ (**Please see the manuscript, page 25-27, line 317-349**) and revised the manuscript to strengthen the above novelty. Moreover, our results present not only reaction mechanism but comprehensively indicate what is needed to achieve low-temperature NH_3 -SCR in a practical condition from multiple points of views such as material, structure, kinetics and spectroscopy. As reviewers #1, #2 and #3 suggested, there are some points that we have to deal with and improve to be a more valuable manuscript. We refined the manuscript so that it is suitable for publication in Nature Communication.

Remarks:

Q1. *What are the different roles of Brønsted and Lewis acid sites in the NH_3 -SCR in terms of reaction rate and pathway?*

A1. A previous study on a supported V-based catalyst reported that the reaction rate of Lewis acid mechanism is faster than that of Brønsted acid mechanism (*J. Am. Chem. Soc.* 2017, 139, 44, 15624–15627). As reviewer #3 suggested, although Brønsted acid sites are not directly involved in the catalytic cycle, they play a role in an NH_3 pool to supply NH_3 to the Lewis acid sites located next to the redox V^{5+} sites in the case of supported V-based catalysts. However, our results showed that Brønsted acid sites can directly work as active sites, which is not similar to previous insights. Under a wet condition, NH_3 species adsorbed on Lewis acid sites were dominantly confirmed from *operando* IR measurement for a supported catalyst ($V-W/TiO_2$, **Figure 10**). On the other hand, NH_4^+ species adsorbed on Brønsted acid sites were exclusively seen for bulk W-substituted vanadium oxide catalysts. The result indicates that Brønsted acid sites can be directly involved in the catalytic cycle for bulk W-substituted vanadium oxide catalysts. In general, Lewis acid is deactivated by water. Therefore, it is thought that bulk W-substituted vanadium oxide catalyst is active at a low temperature and in the presence of water because Brønsted acid sites are located next to redox active sites.

Q2. *It was stated in the manuscript text that “The role of water is thought to inhibit the*

adsorption by influencing adsorption of the reactant and/or the reaction between NO and adsorbed NH₃". Either strong experimental evidence or references should be provided to support this mechanism on the effect of water in SCR.

A2. We should have added evidence or references to the sentence. Previous works provided the kinetic evidence to the effect of water on reaction cycle. The following papers were added as references to the sentence that reviewer 2 pointed out (**Page6, line77-78**).

37. Lintz, H. G. & Turek, T. Intrinsic kinetics of nitric oxide reduction by ammonia on a vanadia-titania catalyst. *Appl. Catal. A, Gen.* **85**, 13–25 (1992).
38. Dumesic, J. A., Topsøe, N. Y., Topsøe, H., Chen, Y. & Slabiak, T. Kinetics of selective catalytic reduction of nitric oxide by ammonia over vanadia/titania. *J. Catal.* **163**, 409–417 (1996).
39. Lietti, L., Nova, I. & Forzatti, P. Selective catalytic reduction (SCR) of NO by NH₃ over TiO₂-supported V₂O₅-WO₃ and V₂O₅-MoO₃ catalysts. *Top. Catal.* **11–12**, 111–122 (2000).
40. Turco, M., Lisi, L., Pirone, R. & Ciambelli, P. Effect of water on the kinetics of nitric oxide reduction over a high-surface-area V₂O₅/TiO₂ catalyst. *Appl. Catal. B, Environ.* **3**, 133–149 (1994).
41. Tufano, V. & Turco, M. Kinetic modeling of nitric oxide reduction over a high-surface area V₂O₅-TiO₂ catalyst. *Appl. Catal. B, Environ.* **2**, 9–26 (1993).
42. Long, R. Q. & Yang, R. T. Selective catalytic reduction of NO with ammonia over V₂O₅ doped TiO₂ pillared clay catalysts. *Appl. Catal. B Environ.* **24**, 13–21 (2000).
43. Forzatti, P. Present status and perspectives in de-NO_x SCR catalysis. *Appl. Catal. A Gen.* **222**, 221–236 (2001).
44. Forzatti, P. Environmental catalysis for stationary applications. *Catal. Today* **62**, 51–65 (2000).
45. Topsøe, N.-Y., Slabiak, T., Clausen, B. S., Srnak, T. Z. & Dumesic, J. A. Influence of water on the reactivity of vanadia/titania for catalytic reduction of NO_x. *J. Catal.* **134**, 742–746 (1992).
46. Nova, I., Lietti, L., Tronconi, E. & Forzatti, P. Transient response method applied to the kinetic analysis of the DeNO_x-SCR reaction. *Chem. Eng. Sci.* **56**, 1229–1237 (2001).

Q3. *What are crystal structure and morphology of samples loaded with different W content after reaction in the presence of water?*

A3. XRD patterns and SEM images of 0W-V (without tungsten) and 3.5W-V (with tungsten) were additionally measured before and after the activity test. XRD patterns were the same before and after the stability test. For SEM measurements, the morphology of 3.5W-V was not changed after the stability test. On the other hand, 0W-V had a pore-like structure but the morphology changed to a block-like smooth structure after the measurements. Thus, the structure is maintained by the introduction of tungsten. The following corrections were made.

- SEM images of 0W-V and 3.5W-V after the stability test were added to the supplementary information as the **supplementary figure 6 (PageS12)**.
- XRD patterns of 0W-V and 3.5W-V after the stability test were added to the supplementary information as the **supplementary figure 7 (PageS13)**.
- The description of crystal structures and morphology of 0W-V and 3.5W-V was added to the main text (**Page14, line184-188**).

Q4. *In the presence of water, water is dissociatively adsorbed on the W-V oxide. The hydroxyl groups are serving as Brønsted acid site. The remaining question on the role of proton or hydride must be addressed.*

A4. The remaining proton can also form Brønsted acid sites as shown in Figure 8g in the manuscript. It has previously reported that Lewis acid sites (metal cations) and Brønsted base sites (oxygen ions) can be changed into Brønsted acid sites by dissociative adsorption of water (ref. 58). OH groups form on coordinatively unsaturated W metal cations as we already described. Remaining protons react with bridging oxygen, forming new Brønsted acid sites. The following corrections were made.

- The description of the formation of Brønsted acid sites by water was modified (**Page22, line289-291 and page223, line 293**)
- The following reference was added for the explanation of dissociative adsorption of water:
58. H.H. Kung. *Transition Metal Oxides: Surface Chemistry and Catalysis*. (Elsevier, 1989).

Response to reviewer #3:

We appreciate reviewer 3 for careful and profound reading of this manuscript and the helpful comments and useful suggestions for further improvement of this manuscript. We have answered your points below.

Comments:

In this manuscript, the low-temperature NH₃-SCR reaction mechanism over W-substituted vanadium oxide catalysts in the presence of water was investigated by various experimental characterizations. The authors found that the 3.5 mol% W-substituted vanadium oxide showed excellent NH₃-SCR performance at low temperature, which was attributed to the fact that the moderate tungsten loading facilitated the formation of Brønsted acid sites in the presence of water, and maintained high redox ability of the vanadium oxide in the meanwhile. The finding is interesting and is important for the development of SCR catalyst, however, the quality of the manuscript should be substantially improved before the reconsideration. Further clarification and correction for the issues listed below is required.

Response:

We would like to thank the reviewer for the positive feedback and thoughtful comments.

Remarks:

Q1. *The SCR activity test was carried out under 10 vol.% water (Figure 3), while the operando IR and UV-Vis measurements (Figure 8 and 9) were performed under 2 vol.% water. Why didn't the authors use the same humidity conditions? What are the IR and UV-Vis results under 10 vol.% water?*

A1. As reviewer #3 suggested, the *operando* experiments should be conducted under 10 vol.% water, which is the same condition as activity test, to ensure consistency. However, water vapor affected S/N ratio of IR and MS spectra and satisfied data could not be obtained under high water vapor volume such as 10%. The activity test in the presence of 2% water was conducted to make sure that the effect of water is the same as that in the *operando* measurements. The NO conversion in the presence of 2% and 10% water was similar to the values under 5-20 vol.% water. Thus, the effect of water would be the same under both conditions and the result of *operando* measurements can explain the result obtained from the activity test in the presence of water. The following corrections were made.

- The NO conversion of 0W-V (without tungsten) and 3.5W-V (with tungsten) in the presence of 2% was additionally plotted in **Figure 4a (Page13)**.
- The description of the effect of water was corrected (**Page13, line172-176**).

Q2. *Although the author claims that the prepared vanadium oxide catalyst with bulk crystal structure has low oxidation ability of SO₂, this point is doubtful. The author should show the SCR of NO_x activity with the presence of SO₂, and the conversion of SO₂.*

A2. As for tolerance to SO₂ and SO₂ oxidation ability on bulk vanadium oxide-based catalysts, we have already reported the SO₂ conversion and NH₃-SCR activity in the presence of SO₂ (*ACS Catal.*, 9, 9327–9331, 2019, figureS12 and S13). It was experimentally confirmed that oxidation of SO₂ does not occur at a low temperature such as 150°C. Although we are aiming for the catalyst working at 100-150°C, bulk vanadium oxide might proceed with SO₂ oxidation reaction at a much higher temperature like >250-300°C because of high vanadium content as reviewer #3 concerns (*Catalysis Letters*, 73, 79–83, 2001).

Q3. *The authors claim that the apparent activation energy (E_a) for the NH₃-SCR over the 0W-V catalyst in the presence of water (22 kJ mol⁻¹) is lower than that without water (39 kJ mol⁻¹, Line 213 and Table 1). However, the NH₃-SCR activity in the presence of water is lower than that under dry condition (Figure 3). The two results are contradictory.*

A3. As reviewer #3 mentioned, the apparent activation energy of 0W-V (only V₂O₅) was lower than that of 3.5W-V (with tungsten) although 3.5W-V showed the larger activity or reaction rate. If we focus on the preexponential factor (intercept of the lines), the value for 0W-V is smaller than that for 3.5W-V. The preexponential factor is regarded as the constant depending on the number of active sites. Therefore, we can understand that the reaction mechanism of 0W-V can change to the path with smaller apparent activation energy but the number of active sites decreases because the active sites are blocked by water. Thus, NO conversion and reaction rate of 0W-V are smaller than those of 3.5W-V. Water would affect each primary step in the NH₃-SCR cycle. Therefore, the elucidation would be the future task. The following addition was made.

- The explanation about the preexponential factor was added to the manuscript and the

description about Arrhenius plot was updated (**Page16, line204-209**).

Q4. As shown in Figure 9d, the amounts of N₂ production under wet condition seems to be higher than that under dry condition, instead of “almost the same” (line 313 of page 26), which is also contradicts the activity results (Figure 3).

A4. As reviewer #3 suggested, the amounts of N₂ produced under a wet condition seems to be slightly larger than that under a dry condition. The activity test was conducted under steady state while the *Operando* measurement was performed under a transient state. Therefore, there would be a difference in the results obtained from these two measurements. Although there is such difference, it can be concluded from the data in figure 9d that water does not inhibit the production of nitrogen for 3.5W-V under a transient condition. We corrected the description of this data instead of “almost the same”. Also, NO+NH₃ (i.e. without oxygen) flow to the sample in the case of *operando* measurements but there might be the diffusion of oxygen ion from bulk inner part to the surface, which is not like the supported V₂O₅/TiO₂ catalyst. Therefore, V sites reduced by NO+NH₃ might be re-oxidized by bulk lattice oxygen and there would be the chance that the reaction of NO+NH₃ proceeds again on re-oxidized redox sites and acid sites. It is speculated that the production of N₂ under a wet condition is higher than that under a dry condition because the diffusion of oxygen ion might be facilitated in the presence of water. As for these points, the following correction was made.

- The explanation of Figure 9d was corrected as below (**Page25, line312-315**):
Notably, considerable N₂ production was seen even in the presence of water (*Figure 9d, mass spectra*). These results demonstrate that 3.5W-V proceeds with NH₃-SCR by the redox cycle and the N₂ production is not affected by water in a transient state.

Q5. In some recent important progresses of SCR research (*Angew. Chem. Int. Ed.* 2016, 55, 11989; *Sci. Adv.* 2018, 4, eaau4637), Lewis acid sites have been demonstrated to be the major active sites, while Brønsted acid sites are not directly involved in the catalytic reaction and mainly serve as an NH₃ pool to replenish the Lewis acid sites. This study shows that the NH₃-SCR activity under dry condition is higher than that in the presence of water and the excess tungsten substitution decreases the activity (Figure 3), which are consistent with the above conclusion. However, the authors also emphasized that importance of Brønsted acid

sites. These understandings need to be unified.

A5. As reviewer #3 mentioned, the recent consensus for V_2O_5/TiO_2 is that Lewis acid sites mainly act as active sites and Brønsted acid sites serve as a supplier of ammonia to Lewis acid sites. However, Brønsted acid sites were mainly confirmed for bulk W-substituted vanadium oxide catalysts in this work, especially in the presence of water, which is not similar to a supported V catalyst. To consider the difference between supported and bulk catalyst, *operando* IR measurements were additionally conducted at 200 °C for 3.5W-V and V-W/ TiO_2 in the presence of water under NO flowing (**Figure 10 in manuscript**). After the adsorption of NH_3 under a wet condition, NH_4^+ species adsorbed on Brønsted acid sites were solely confirmed for 3.5W-V while NH_3 species adsorbed on Lewis acid sites were mainly observed for V-W/ TiO_2 . In the case of 3.5W-V, NH_4^+ species were rapidly consumed and significant N_2 production was confirmed when NO flowed. On the other hand, the reactivity of Lewis acid sites and Brønsted acid sites of V-W/ TiO_2 was low and N_2 production was less. These results indicate that bulk W-substituted vanadium oxide catalysts can directly use Brønsted acid sites and the reactivity of Brønsted acid sites of the bulk W-substituted vanadium oxide is higher than those of Brønsted acid and Lewis acid sites of a conventional V-W/ TiO_2 catalyst. As supplementary measurements, another *operando* IR measurements and *operando* UV-Vis measurements were also conducted to support the above data (**Supplementary Figure 14 and Supplementary Figure 15**). Lewis acid site is generally deactivated by water but Brønsted acid site is not. In terms of water tolerance, Brønsted acid sites would be the crucial factors to achieve low temperature NH_3 -SCR under a wet condition. Thus, we emphasized the importance of Brønsted acid sites in the manuscript. The following additions were made.

- *Operando* IR and UV-Vis measurements were additionally conducted at 200 °C for 3.5W-V and V-W/ TiO_2 in the presence of water (**Page27, Figure 10 in the main text and Page S22-23, Supplementary Figure 14 and 15**).
- The description of the difference between bulk and supported catalyst was added to the main text (**Page25-27, line 317-349**).

Q6. There are many spelling errors. For instance, in the caption of Figure 1, the second “(b)” and “(c)” should be “(d)” and “(e)”, respectively, and the “Enralrged” should be “Enlarged”. In the line 133 of page 10, the “obserbed” should be “observed”. In the caption of Figure 2, the “Enralrged” should be “Enlarged”, and the “010” should be corrected to “(010)”. In Figure 7b,

the “ΔKM300” should be corrected to “ΔKM700. In the caption of Figure 8, the “(e) Brønsted acid and (f) Lewis acid” should be corrected to “(e) Lewis acid and (f) Brønsted acid”. In the line 143 and 288, the “figure” should be “Figure”. Also, the language should be substantially improved with professional assistance.

A6. We would like to thank reviewer #3 for suggesting errors in detail. They should have been carefully checked. The errors shown above were corrected and another error was also checked.

REVIEWERS' COMMENTS

Reviewer #2 (Remarks to the Author):

The authors have carefully and appropriately addressed all of my critiques and suggestions. Acceptance to publication is recommended.

Reviewer #3 (Remarks to the Author):

I maintain my judgment on the novelty and importance of the revised paper, that is, the finding in this manuscript is interesting and is important for the development of SCR catalyst. I am satisfied that all my questions have been properly answered by the author and the paper has also been made the corresponding revision. Therefore I recommend publishing this paper now.